# Radical Scavenging-Linked Anti-Adipogenic Activity of *Aster scaber* Ethanolic Extract and Its Bioactive Compound

**DOI:** 10.3390/antiox9121290

**Published:** 2020-12-16

**Authors:** Ye-Eun Choi, Sun-Il Choi, Xionggao Han, Xiao Men, Gill-Woong Jang, Hee-Yeon Kwon, Seong-Ran Kang, Jin-Soo Han, OK-Hwan Lee

**Affiliations:** 1Department of Food Science and Biotechnology, Kangwon National University, Chuncheon 24341, Korea; ye0538@hanmail.net (Y.-E.C.); docgotack89@hanmail.net (S.-I.C.); xionggao414@hotmail.com (X.H.); menxiaodonglei@naver.com (X.M.); jkw5235@naver.com (G.-W.J.); sosakwon1@naver.com (H.-Y.K.); 2The Food Industry Promotional Agency of Korea, Iksan 54576, Korea; srkang0817@foodpoils.kr (S.-R.K.); jinsoo07@foodpoils.kr (J.-S.H.)

**Keywords:** anti-oxidant, reactive oxygen species, anti-obesity, 3T3-L1, *Aster scaber*

## Abstract

*Aster scaber* is a wild vegetable cultivated in Korea and is known to contain phytochemicals with various biological activities. The potential antioxidant and anti-obesity effects of *A. scaber* and their mechanism are yet to be reported. We evaluated the total phenolic, flavonoid, and proanthocyanidin contents and oxygen radical absorbance capacity of *A. scaber* ethanolic extract (ASE), and analyzed the major phenolic compounds of ASE. Antioxidant activity was measured at the chemical level through 2,2-diphenyl-1-picrylhydrazyl (DPPH), reducing power assay, 2,2′-azino-bis(3-ethylbenzthiazoline-6-sulfonic acid) (ABTS), and fluorescence recovery after photobleaching (FRAP) assay. In addition, it was measured in vitro through inhibition of Reactive oxygen species (ROS) production in 3T3-L1 adipocyte, and inhibition of lipid accumulation was also evaluated. ASE reduced the expression of enzymes involved in the production of ROS and increased the expression of antioxidant enzymes that reduce increased ROS levels. They also reduced the expression of adipogenesis transcription factors that regulate adipocyte differentiation in relation to ROS production, inhibited the expression of lipogenesis-related genes related to fat accumulation through AMP-activated protein kinase (AMPK) activity, and increased expression of lipolysis-related genes. Thus, ASE containing CGA (chlorogenic acid) inhibits ROS production in 3T3-L1 adipocytes, owing to its strong antioxidant activity, and inhibits lipid accumulation caused by oxidative stress. The extract can be used as a potential functional food material for reducing oxidative stress and obesity.

## 1. Introduction

The increasing prevalence of obesity, which indicates excessive fat accumulation in the body, is steadily progressing globally. Obesity is caused by an increased energy supply in relation to the readily available food supply in high-income countries, combined with decreased physical activity [1,2,3]. Obesity is often accompanied by the onset of chronic metabolic diseases, including insulin resistance, type 2 diabetes, and cardiovascular diseases, which cause a wide range of complications. This suggests that adipose tissue functions as an endocrine organ secreting adipokines, including adiponectin and leptin; accumulating triglycerides; and acting as a source of free fatty acids [4]. Adipose tissue growth can be divided into adipocyte hyperplasia and hypertrophy. The former increases the number of adipocytes for triglyceride accumulation through adipocyte differentiation. Dilation of adipose tissue causes increased invasion of macrophages and increases inflammation, oxidative stress, and insulin resistance [5,6,7]. During adipocyte adipogenesis, tissues exhibit increased oxidative stress through the activation of the Reactive oxygen species (ROS) production system and inactivation of antioxidant enzymes. Oxidative stress in mature adipocytes is a major cause of obesity and is induced by adipogenesis promotion, insulin resistance, type 2 diabetes, and metabolic disease induction, such as atherosclerosis, by inhibiting glucose uptake. Therefore, it is important to regulate the mechanisms associated with the inhibition of ROS production and lipid accumulation in adipocytes [8,9,10].

ROS are representative oxide radicals that cause oxidative stress in cells and are produced through the oxidation of nutrients during energy generation in body tissues and cells. ROS production is closely related to lipid accumulation in adipocytes, and excessive ROS production can cause diabetes and other complications [11]. According to recent reports, fat accumulation increases the production of ROS, which is associated with increased expression of Nicotinamide Adenine Dinucleotide Phosphate (NADPH) oxidase [7]. Glucose-6-phosphate dehydrogenase (G6PDH) and NADPH oxidase 4 (NOX4) regulate the production of ROS in adipocytes. Glucose acts as a nutrient source during adipocyte differentiation and is involved in an energy metabolism pathway related to fat synthesis for regulating glucose entering the cell. Glucose metabolism refers to the process by which glucose is oxidized to CO_2_ through glycolysis and the citric acid (TCA) cycle. G6PDH is an NADPH-generating enzyme that breaks down glucose in the cytoplasm by a hexose monophosphate (HMP) shunt to produce NADPH, which is necessary for fat synthesis. The generated NADPH is converted to NADP^+^ by NOX4, and ROS such as superoxide (O_2_^−^) are generated during the process. ROS produced in mitochondria is a superoxide (O_2_^−^), and its synthesis increases when mitochondria fail to produce ATP and maintain a high NADPH/NAD^+^ ratio [12].

Excessive ROS are removed by antioxidant enzymes, which are classified as non-enzymatic and enzymatic antioxidants. Non-enzymatic antioxidants work by interfering with the chain reaction of free radicals. The antioxidant glutathione naturally exists in two forms: Reduced glutathione (GSH) and oxidized glutathione (GSSG). GSH is converted to GSSG by the action of reducing equivalents to produce other unstable molecules such as ROS. The antioxidant glutathione confers electrons to glutathione peroxidase (GPx) and glutathione reductase (GR), which are classified as enzymatic antioxidants. GPx can use GSH as a reducing agent to catalyze the conversion of hydrogen peroxide (H_2_O_2_) into water. GR increases GSH levels by catalyzing the conversion of GSSG to GSH and indirectly performs glutathione’s antioxidant action. Enzymatic antioxidants work by decomposing and neutralizing free radicals. These antioxidant enzymes require trace amounts of metal cofactors (copper, zinc, manganese, and iron) and convert free radicals first to hydrogen peroxide (H_2_O_2_) and then to water to scavenge ROS. Superoxide dismutase (SOD) exists as Cu/Zn-SOD when present in the cytoplasm and as Mn-SOD when present in the mitochondrial matrix. SOD catalyzes the conversion of superoxide (O_2_^−^) to O_2_ and H_2_O_2_ and serves as the first line of defense against oxygen-derived free radicals. Catalase completes the detoxification process initiated by SOD by converting H_2_O_2_ into O_2_ and H_2_O using iron and manganese as cofactors [13,14,15].

Recent studies have shown a link between ROS production and adipocyte differentiation. Increasing intracellular ROS levels induce adipocyte differentiation by promoting the mitotic clonal expansion phase, an important step in adipocyte differentiation [16,17]. Adipocyte differentiation initiates preadipocyte differentiation through the expression of CCAAT/enhancer-binding protein (C/EBPα) and C/EBP β and peroxisome proliferator-activated receptor γ (PPARγ). The activity of PPARγ increases insulin sensitivity and adipogenesis [17,18]. Fatty acid-binding protein (aP2) and fatty acid synthase (FAS) are expressed by C/EBPa and PPARγ and play important roles as mediators of fatty acid synthesis during adipogenesis [19].

*Aster scaber* is a perennial herb belonging to the Asteraceae family that grows and is cultivated in the mountainous areas of Korea. It is rich in nutrients such as vitamin C, Ca, Fe, and β-carotene [20]. *A. scaber* leaves are known to contain caffeoylquinic acid (CQA) compounds, flavonoids, and terpenoids [21,22]. Moreover, its flowers and stems contain essential oils and its roots contain functional ingredients such as coumarin, sapogenin, shionon, alkaloid, squalene, and friedelin [23]. In addition, phenolic compounds of various CQAs have been isolated from *A. scaber* leaves [24], and one of these compounds, ((-)-3,5-dicaffeoyl-mono-quinic acid), has been reported to inhibit integrase, the enzyme necessary for the multiplication of human immunodeficiency virus [25,26]. Several studies on the functionality of *A. scaber* have reported its antioxidant and anti-obesity efficacy of five wild herb extracts; of them, *A. scaber* had the highest total phenolic content and flavonoid content and had an excellent anti-obesity efficacy [27]. Thus, owing to these varied functions of *A. scaber*, data supporting the use of this herb as a functional health food are required.

We particularly assessed *A. scaber* in this study because it has been demonstrated to have high antioxidant activity and excellent lipid accumulation inhibitory effect in 3T3-L1 adipocyte [27]. Here, we identified and quantitatively analyzed the phenolic compounds of *A. scaber* and assessed its antioxidant and anti-obesity activities in 3T3-L1 adipocytes. The inhibitory activities of *A. scaber* against ROS production and lipid accumulation in adipocytes was confirmed using 3T3-L1 cells, and the mechanism underlying this activity was investigated.

## 2. Materials and Methods

### 2.1. Chemicals

High-performance liquid chromatography (HPLC)-grade methanol and acetonitrile were procured from J.T. Baker (NJ, Phillipsburg, USA). Ascorbic acid, gallic acid, Folin–Ciocalteu phenol reagent, potassium ferricyanide, potassium persulfate, 6-hydroxy-2,5,7,8-tetramethylchroman-2-carboxylic acid (Trolox), catechin, acetic acid, formic acid, vanillin, 2,2-azobis(2-amidino propane) dihydrochloride (AAPH), fluorescein sodium salt (FL), sodium nitrite (NaNO_2_), sodium carbonate, and chlorogenic acid were obtained from Sigma (St. Louis, MO, USA). 2,2-diphenyl-1-picrylhydrazyl (DPPH), 2,2′-azino-bis(3-ethylbenzthiazoline-6-sulfonic acid) (ABTS), N-acetyl-L-cysteine (NAC), potassium ferricyanide, fluorescein sodium salt (FL), 2,4,6-tris(2-pyridyl)-s-triazine, acetic acid, 2,2-azobis(2-amidino propane) dihydrochloride, Folin–Ciocalteu phenol reagent, sodium nitrite, insulin, and Oil Red O were supplied by Sigma Aldrich Co. (St. Louis, MO, USA). Fluorescein sodium salt and potassium phosphate monobasic, dibasic anhydrous were obtained from Junsei Chemical (Tokyo, Japan). Trypsin-ethylenediaminetetraacetic acid (Trypsin-EDTA, 0.05%), bovine serum, Dulbecco’s modified eagle’s medium powder (DMEM), fetal bovine serum, penicillin-streptomycin, and phosphate-buffered saline were obtained from Gibco (Gaithersburg, MD, USA). The primary and secondary antibodies against PPARγ, C/EBPα, aP2, NOX4, G6PDH, SOD1, SOD2, GR, GPX, CAT, FAS, HSL, phosphorylated HSL, AMPK, p-AMPKα, ACC, p-ACC, and GAPDH were procured from Cell Signaling Technology (Danvers, MA, USA) and Santa Cruz Biotechnology (Dallas, TX, USA).

### 2.2. Preparation of A. scaber Ethanolic Extract

*A. scaber* harvested in 2019 was purchased from the Yangyang market (Gangwon, Korea), and *A. scaber* ethanolic extract (ASE) was obtained using the following procedure. *Aster scaber* herb (500 g) was freeze-dried using a commercial freeze dryer (Ilshin, Seoul, Korea) and then coarsely pulverized through a 20–30 mesh using a grinder (IKAM20, IKA, Staufen, Germany). The obtained dry *A. scaber* powder (30 g) was mixed with 70% ethanol (600 mL), and the components were extracted for 3 h at 80 °C. The sample was filtered using Whatman No. 2 filter paper (Whatman Ltd., Maidstone, Kent, UK) and concentrated under reduced pressure at 50 °C using a vacuum evaporator (Eyela, Tokyo, Japan). The extract was then freeze-dried (Ilshin, Seoul, Korea).

### 2.3. Total Phenolic, Flavonoid, Proanthocyanidin Contents, and Oxygen Radical Absorbance Capacity Value

The total phenolic content was determined according to a modified Folin–Ciocalteu et al. [28]. The sample was dissolved in 70% ethanol solution (1 mL) in a test tube and mixed with 10% Folin–Ciocalteu reagent (1 mL) and 2% sodium carbonate solution (1 mL). After allowing the mixture to react in the dark for 1 h, using a microplate reader, the absorbance was measured at 750 nm (Molecular Devices, Sunnyvale, CA, USA). The calibration curve was plotted using various concentrations of gallic acid, and the result was calculated as using the equation y = 13.642x − 0.0864 and expressed as milligrams of gallic acid equivalents per gram.

Total flavonoid content was determined according to the method reported by Zhishen et al. [29] with modifications. Samples dissolved in 70% ethanol solution (0.5 mL), 95% ethanol (1.5 mL), 1 M potassium acetate (0.1 mL), and 10% aluminum nitrate (0.1 mL) were added to a test tube, and then mixed with distilled water (2.8 mL). After reacting for 30 min at 20–22 °C, using a microplate reader, the absorbance was measured at 415 nm (Molecular Devices, Sunnyvale, CA, USA). Quercetin was used as the standard for plotting the calibration curve. Results were calculated using the equation y = 3.5469x − 0.0491 and expressed as milligrams of gallic acid equivalents per gram.

Proanthocyanidin content was determined according to the modified vanillin-hydrochloric acid method [29]. Samples dissolved in methanol (0.1 mL) and 0.5% vanillin-HCl solution (0.5% vanillin [*w*/*v*] + 4% HCl in methanol) were added to test tubes and placed in a dark place at 20–22 °C for 20 min. Then, using a microplate reader, the absorbance was measured at 500 nm (Molecular Devices, Sunnyvale, CA, USA). The proanthocyanidin content was calculated by plotting a standard curve for catechins (0.03125–0.5 mg/mL). Results were calculated using the equation y = 0.2351x − 0.0385 and expressed as milligram catechin equivalents per gram.

The oxygen radical absorbance capacity (ORAC) assay was performed using the modified method reported by Ou et al. [30]. For this, the sample was diluted using 75 mM sodium potassium phosphate buffer. The diluted samples (25 μL) combined with fluorescein (150 μL) were added to a black-welled 96-well plate. Subsequently, 18 mM APPH (25 μL) was transferred to each well and was incubated at 37 °C for 15 min. The resulting fluorescence was measured using a micro plate reader (Molecular Devices, Sunnyvale, CA, USA). We recorded excitation wavelength of 485 nm and emission wavelength of 530 nm every 3 min for 90 min at 37 °C. The ORAC value was calculated using Trolox’s calibration curve (0.3125–10 mg/mL) and was expressed as micromoles of Trolox equivalent per gram.

### 2.4. HPLC-Tandem Mass Spectrometry (MS)

To identify phenolic compounds in ASE, a CAPCELL PAK C18 UG120 S5 (250 × 4.6 mm, 5 μm; OSAKA SODA, Osaka, Japan) column was used in a 1260 Infinity HPLC system (Agilent Technologies, Santa Clara, CA, USA) connected to a 6550 Q-TOF mass spectrometer (Agilent Technologies, Santa Clara, CA, USA). The representative wavelength of detection was 272 nm, the flow rate was 0.7 mL/min, and the column oven temperature was 28 °C. The mobile phase comprising solvent A (water:acetic acid = 99:1, *v*/*v*) and solvent B (acetonitrile) was isocratically eluted (Appendix A). The quadrupole time-of-flight mass spectrometer was operated using N_2_ as the drying gas at a flow rate was 17 L/min, a temperature of 225 °C, a nebulizer pressure of 45 psi, sheath gas flow of 11 L/min., and a sheath gas temperature of 350 °C. In addition, the capillary voltage was 3500 V, the nozzle voltage was 2000 V, the fragmentor voltage was 150 V, the skimmer voltage was 65 V, the OCT 1RF Vpp was 750 V, and the collision energy was 25 V. Mass pattern analysis was conducted by operating the mass spectrometer in the negative ion mode (Appendix A).

### 2.5. HPLC-Photodiode Array

The analysis of chlorogenic acid was performed by Roby et al. [31]. Chlorogenic acid content in ASE was analyzed by HPLC using Waters 2695 Separation Module instrument and 996 Photodiode Array Detector (Milford, MA, USA). The analytical column used was Capcell Pak C18 MG (Shiseido, 4.6 × 250 mm, 5.0 μm, Tokyo, Japan). ASE and chlorogenic acid were dissolved in methanol and filtered through a 0.45-μm pore size polyethylene terephthalate (PET) syringe filter (Whatman, Maidstone, United Kingdom, USA) before injecting in HPLC instrument. The mobile phase consisted of 0.05% formic acid in water and methanol, and the sample (50 µL) was eluted isocratically at room temperature under a flow rate of 0.8 mL/min. The Photodiode Array (PDA) detector was set to a detection wavelength of 280 nm (Appendix A).

### 2.6. Antioxidant Activity Assays

The DPPH assay was performed by adding 0.2 mL of sample and 0.4 mM DPPH solution (0.8 mL) to a test tube. After mixing the solution and allowing it to react for 10 min in dark at 20–22 °C, using a microplate reader, the absorbance was measured at 517 nm (Molecular Devices, Sunnyvale, CA, USA). The blank was prepared with distilled water, and ascorbic acid was used as a positive control. All experiments were repeated three times. The radical scavenging activity was calculated using the following equation:DPPH radical scavenging activity (%) = [1 − (A_sample_/A_blank_)] × 100

The reducing power assay was performed according to the method reported by Oyaizu [32]. In test tubes, samples (0.1 mL) of various concentrations were mixed with potassium ferricyanide (1%, 0.5 mL) and sodium phosphate buffer (0.2 M, 0.5 mL) and incubated at 50 °C for 20 min. Trichloroacetic acid (0.5 mL) was then added to the mixed solution and centrifuged at 1790× *g* for 10 min. In a new tube, the supernatant (0.5 mL), iron (III) chloride solution (0.1%, 0.1 mL), and distilled water (0.5 mL) were mixed. All experiments were repeated three times. Using a microplate reader, the absorbance of this solution was measured at 700 nm (Molecular Devices, Sunnyvale, CA, USA).

The ABTS assay was based on the method reported by Re et al. [33] with some modifications. ABTS (7 mM, *w*/*v*) was mixed in water and potassium persulfate (2.45 mM, *w*/*v*) and incubated at 20–22 °C for 16 h. The ethanol and ABTS solution was diluted until an absorbance of 0.7 ± 0.02. Then, this ABTS^+^ solution (1 mL) was added to the sample solution (10 μL) of different concentrations. After mixing the solution and allowing it to react for 6 min at 20–22 °C, using a microplate reader, the absorbance was measured at 734 nm (Molecular Devices, Sunnyvale, CA, USA). All experiments were repeated three times. The radical scavenging activity was calculated using the following equation:ABTS radical scavenging activity (%) = [1 − (A_sample_/A_blank_)] × 100

The fluorescence recovery after photobleaching (FRAP) assay was conducted in accordance with the method reported by Benzie and Strain [34]. The FRAP solution was mixed with acetate buffer (pH 3.6, 300 mM), 2,4,6-tri(2-pyridyl)-1,3,5-triazine (TPTZ) solution (10 mM, in 40 mM HCl) and FeCl_3_·6H_2_O solution (20 mM) at a ratio of 10:1:1 (*v*/*v*). The FRAP reagent was prepared freshly for each experiment and was used after incubation at 37 °C for 4 min. Then, the sample (50 μL) was mixed with distilled water (150 μL) and incubated at 37 °C for 4 min. All experiments were repeated three times. Subsequently, using a microplate reader, the absorbance was measured at 593 nm (Molecular Devices, Sunnyvale, CA, USA).

### 2.7. Cell Culture

3T3-L1 preadipocytes were procured from American Type Culture Collection (Manassas, VA, USA). Culture and differentiation medium were cultured in DMEM in a 5% CO_2_ incubator at 37 °C with 10% calf serum and 1% penicillin-streptomycin. After reaching confluency, the cells were allowed to stand for 48 h and then allowed to differentiate for 48 h following treatment with MDI cocktail (0.5 mM 3-isobutyl-1-methylxanthine, 1 μg/mL insulin, and 1 μM dexamethasone), 10% fetal bovine serum, and DMEM containing 1% penicillin-streptomycin. After that, every 2 days, 1 μg/mL insulin was added to DMEM containing 10% FBS and 1% penicillin-streptomycin to change the medium. ASE (50, 100 μg/mL), chlorogenic acid (CGA) (21, 210 μM), and NAC (10 mM) were treated together after 48 h in a confluent state.

### 2.8. XTT Assay

Cell viability was measured using the XTT assay. For this, 3T3-L1 preadipocytes were seeded in a 96-well plate at a density of 3 × 10^5^ cells/well and differentiated following treatment with 0 to 100 μg/mL ASE and CGA for 6 days. The XTT and PMS reagents were mixed and placed in a medium and incubated at 37 °C for 4 h. All experiments were repeated six times. The absorbance was measured at 734 nm.
Cell viability (%) = (A_sample_/A_blank_) × 100

### 2.9. Oil Red O Staining and NBT Assay

Oil Red O is used to stain triglycerides in adipocytes to determine the degree of triglyceride accumulation. After differentiating 3T3-L1 preadipocytes for 6 days, the medium was removed. The cells were then washed with phosphate-buffered saline and fixed with 10% formaldehyde solution, washed with 60% isopropanol, and then dried completely. Then, Oil Red O staining was performed at 20–22 °C for at least 30 min, followed by washing with distilled water and drying. Isopropanol was added to the dried plate to elute the dyed lipid droplets, and the absorbance was then measured at 490 nm to quantify of triglyceride accumulation. All experiments were repeated six times.

After differentiating 3T3-L1 preadipocytes for 6 days, the medium was removed, washed with PBS, and treated with 0.2% NBT solution, followed by incubation at 37 °C for 90 min. ROS accumulated in adipocytes react with NBT solution to produce formazan. Thus, the formazan formed was eluted by mixing DMSO and KOH solution (7:3), and the absorbance of the mixture was measured at 570 nm. All experiments were repeated six times.

### 2.10. Western Blot Analysis

After ASE (50, 100 μg/mL), CGA (21, 210 μM), and NAC (10 mM) treatment, 3T3-L1 cells were dissolved by adding lysis buffer and centrifuging for 15 min at 17,000× *g* at 4 °C. The protein concentration of the supernatant was quantified using a Bio-Rad protein assay kit standardized with bovine serum albumin. Proteins were subjected to sodium dodecyl sulfate-polyacrylamide gel electrophoresis using a running gel and stacking gel. Proteins isolated by electrophoresis were electrophoresed at 100 V for 90 min using a polyvinylidene difluoride membrane (Bio-Rad) and transfer buffer (20% methanol, 25 mM Tris-HCl, and 192 mM glycine) and then blocked with 5% skim milk. To examine the expression of the primary antibody, it was diluted 1:1000 in 1X Tris-buffered saline containing Tween 20 (TBST) solution, reacted for 24 h, and washed 3 times with 1X TBST. The secondary antibody was reacted for 2 h at 20–22 °C and washed again 3 times with 1X TBST. After applying a color developing reagent of enhanced chemiluminescence (BIONOTE, Hwaseong, Korea) to the membrane, the sample was exposed to a ChemiDoc™ imaging system (Bio-Rad, Hercules, CA, USA), developed, and quantified using Image Lab™esoftware. All experiments were repeated three times.

### 2.11. Statistical Analysis

Experimental results were expressed as the means ± standard deviations of triplicate samples, and values for each test group were obtained using SPSS^TM^ (ver. 12.0, SPSS Inc., Chicago, IL, USA). The significance between each test group was compared using the Duncan multi-range test, and results with *p* < 0.05 were considered statistically significant.

## 3. Results

### 3.1. Total Phenolic, Flavonoid, Proanthocyanin Contents and ORAC Value

Plants contain various phytochemicals, including phenols and flavonoids [35]. It is known that phenolic compounds, which are secondary metabolites of plants, have physiological functions that are mediated by binding phenolic hydroxyl groups with macromolecules such as enzyme proteins [36]. These compounds are widely metabolized in vivo and have been reported to affect physiological aspects such as cell death, inflammation, and antioxidant activity by changing the redox potential [37].

We confirmed total phenolic, flavonoid, and proanthocyanidin contents and ORAC values of *A. scaber* ethanolic extract. As shown in Table 1, total polyphenol content, flavonoid content, and proanthocyanidin contents of *A. scaber* ethanolic extract were 91.84 ± 0.52, 53.39 ± 0.29, and 20.08 ± 0.84 mg/g, respectively, and the ORAC value was 222.69 ± 0.30 μmol/g. Lee et al. [10] found that the total phenolic and flavonoid contents of the *A. scaber* methanol extract were 183.5 ± 4.0 mg GAE/g and 103.9 ± 3.4 mg RE/g, respectively. The ORAC value was 41,638 ± 282 μM TE/g. In Thiruvengadam et al. [38], the total phenolic and flavonoid contents in the *A. scaber* methanol extract were 183.39 ± 5.59 and 3.12 ± 0.09 mg/g, respectively. The difference in content may be affected by the difference in the extraction solvent, the origin of the sample, or the harvest time.

### 3.2. Identification of Phenolic Compounds in ASE by HPLC-Q/TOF-MS

To search for the bioactive compounds contained in ASE, phenolic compounds were analyzed using HPLC-Q/TOF-MS. We used a database (TCM, Tea, Metabolite Library; 2014, Agilent, USA) for identification. The peaks that matched the database were quinic acid and chlorogenic acid. We analyzed the ion fragment data for two peaks with the highest area in the chromatogram (Figure 1A). We estimated peak 1 as quinic acid (Figure 1B) and peak 2 as chlorogenic acid (Figure 1C).

*A. scaber* is known to contain various phenolic caffeoylquinic acid (CQA) compounds [38]. Nugroho et al. [23] conducted simultaneous determination of seven CQAs in chwinamul grown in Korea and reported that the content of 5-O-caffeoylquinic acid (chlorogenic acid) in *A. scaber* was the highest at 9.34 ± 0.11 mg/dry weight g. Therefore, in this study, the bioactive compounds of ASE were set to CGA by referring to the results of phenolic compound identification using HPLC-Q/TOF-MS and previous studies.

### 3.3. Determination of Phenolic Compounds in ASE by HPLC-PDA

Figure 2 shows the HPLC chromatogram of CGA and ASE. By comparing the chromatograms of the standard solution and ASE, the retention time and PDA spectra were confirmed [31]. Nugroho et al. [23] reported a chlorogenic acid content of 9.34 ± 0.11 mg/dry weight g in the methanolic extract of *A. scaber*. These contents were lower and higher, respectively, than the chlorogenic acid content obtained in this study (10.98 ± 0.04 mg/g). The difference in results may be attributed to differences in the origin of the *A. scaber*, extraction solvent, and extraction method.

### 3.4. Antioxidant Activities of ASE and Its Major Compounds Using Different Chemical Assays

Accumulated lipids increase oxidative stress, and ROS produced in adipose tissue play an important role in obesity-related metabolic dysfunction, including insulin resistance [7]. Antioxidant testing models differ in many respects. Therefore, it is difficult to completely compare one method with another. Antioxidant activity should not be determined based on a single antioxidant testing model; in fact, several in vitro testing procedures were used for antioxidant evaluation. Therefore, we measured the antioxidant activity of ASE (0.1, 0.5, and 1 mg/mL) and CGA (21, 105, and 210 μM) at various concentrations.

At room temperature, DPPH is a stable free radical that produces a purple color in ethanol, which turns colorless following treatment by antioxidants. Using DPPH, antioxidants can be evaluated quickly and easily [39]. In our study, DPPH radical scavenging activities were 45.55% to 80.83% for ASE concentrations of 0.1, 0.5, and 1 mg/mL, and 18.51% to 56.05% for CGA concentrations of 21, 105, and 210 μM (Figure 3A).

The reducing power assay shows the reducing power that Fe^3+^/ferricyanide mixture converts to Fe^2+^/ferrous by donating hydrogen in the presence of a reducing agent [40]. Reducing power activities were 8.87% to 18.42% in ASE concentrations of 0.1, 0.5, and 1 mg/mL, and 4.36% to 12.85% in CGA concentrations of 21, 105, and 210 μM (Figure 3B).

The color specific to ABTS^•+^ is produced by the oxidation of ABTS by potassium persulfate. The antioxidant activity of a compound is measured using the decolorization reaction that occurs when the formed ABTS^•+^ is removed by the antioxidant compound [33]. In our study, ABTS radical scavenging activities were 8.87% to 18.42% for ASE concentrations of 0.1 to 1 mg/mL and 4.36% to 12.85% for CGA concentrations of 21, 105, and 210 μM (Figure 3C).

In FRAP analysis, iron 2,4,6-tripyridyl-s-triazine (Fe(III)-TPTZ) is reduced to the colored 2,4,6-tripyridyl-s-triazine (Fe(II)-TPTZ) in the presence of antioxidants [41]. In our study, FRAP activities were 0.13% to 0.44% for ASE concentrations of 0.1, 0.5, and 1 mg/mL, and 0.07% to 0.30% for CGA concentrations of 21, 105, and 210 μM (Figure 3D).

### 3.5. Antioxidant and Anti-Obesity Activities ASE Extract and Its Major Compound in 3T3-L1 Adipocyte

#### 3.5.1. Effect of ASE and CGA on the Cell Viability

As shown in Figure 4, 50 and 100 μg/mL concentrations of ASE and 21 and 210 μM concentrations of CGA were treated, and the extracts at all concentrations showed no cytotoxicity compared to the control group. Therefore, in this study, ASE and CGA were treated in 3T3-L1 preadipocytes at concentrations of 50, 100 μg/mL and 21, 210 μM, respectively, to evaluate the efficacy of reducing lipid accumulation and ROS production.

#### 3.5.2. Effect of ASE and CGA on Lipid Accumulaton and ROS Production

The effects of ASE and CGA on lipid accumulation and ROS production were measured by ORO staining and NBT assay during adipogenesis of 3T3-L1 cells. After 3T3-L1 preadipocyte fusion, 50 and 100 μg/mL concentrations of ASE, and 21 and 210 μM concentrations of CGA were treated after 2 days at 2-day intervals for 6 days. The MDI cocktail was treated for differentiation of preadipocytes, and that differentiation induced the formation of lipid droplets in the cytoplasm.

As shown in Figure 5, lipid accumulation was actively induced in the differentiated 3T3-L1 adipocytes of the control group, and the cells treated with ASE and CGA showed a remarkable decrease in intracellular lipid accumulation compared to the control cells. The amount of lipid accumulation decreased to 58.60 ± 0.05 and 81.33 ± 0.05 in 100 μg/mL ASE and 210 μM CGA, respectively, compared to the negative control group (CON).

The result of measuring dark blue formazan produced by reacting with ROS accumulated in adipocytes is shown in Figure 5. The amount of ROS production decreased to 83.56 ± 0.03 and 89.35 ± 0.04 in 100 μg/mL ASE and 210 μM CGA, respectively, compared to the negative control group (CON).

#### 3.5.3. Effect of ASE and CGA on The Protein Expression of ROS-Generating Factor

The expression levels of glucose-6-phosphate dehydrogenase (G6PDH) and NADPH oxidase 4 (NOX4) were evaluated to determine the effect of ASE and CGA treatment on ROS production (Figure 6). Protein expression of G6PDH and NOX4 was significantly reduced by treatment with ASE and CGA. These results indicate that inhibition of the expression of G6PDH and NOX4 inhibits ROS production during adipogenesis.

#### 3.5.4. Effect of ASE and CGA on The Protein Expression of ROS Regulation Antioxidant Enzyme

The expression levels of Cu/Zn-SOD, Mn-SOD, GPx, GR, and CAT were evaluated to determine the effect of ASE and CGA treatment on ROS regulating antioxidant enzymes. ASE and CGA treatment of adipocytes increased the protein expression of Cu/Zn-SOD, Mn-SOD, CAT, GPX, and GR in a concentration-dependent manner compared to the control group. Treatment with NAC inhibited the protein expression of Cu/Zn-SOD, Mn-SOD, CAT, GPX, and GR, indicating the use of different mechanisms for ROS scavenging. These results show that treatment with ASE and CGA inhibits ROS accumulation by regulating antioxidant enzymes (Figure 7).

#### 3.5.5. Effect of ASE and CGA on the Protein Expression of Adipogenic Transcription Factor

To investigate the effect of ASE and CGA treatment on adipocyte differentiation, the expression levels of C/EBPα, PPARγ, and aP2 were determined. The protein expression of C/EBPα was decreased by ASE and CGA compared to the control group, and the protein expression of PPARγ and aP2 was dose-dependently decreased by ASE and CGA treatment. These results indicate that ASE and CGA treatment inhibited adipogenic differentiation by inhibiting the expression of C/EBPα, PPARγ, and aP2 (Figure 8).

#### 3.5.6. Effect of ASE and CGA on Protein Expression of AMPK Activation and Lipid Metabolism-Related Enzymes

Lipid accumulation in adipocytes can be reduced through inhibition of adipogenesis and increased fat oxidation. Fatty acid synthase (FAS) is a key enzyme for fatty acid synthesis regulation and is expressed in the final stage of adipocyte differentiation [41]. Acetyl-CoA carboxylase (ACC) regulates fatty acid biosynthesis by converting acetyl CoA into malonyl-CoA, while hormone-sensitive lipase (HSL) supplies energy to various tissues through the hydrolysis of triglycerides, and increases the release of fatty acids [42,43]. AMP-activated protein kinase (AMPK) is an enzyme that maintains energy homeostasis and plays an important role in regulating lipid metabolism in adipocytes [44]. AMPK, which is activated by increasing the AMP/ATP ratio, regulates lipid synthesis and fatty acid oxidation through the regulation of FAS, ACC, and HSL [45,46,47].

Gene expression related to lipid production and degradation was investigated to determine the effect of ASE and CGA treatment in adipocytes on the inhibition of lipid accumulation. Protein expression of FAS was decreased through ASE and CGA compared to the control, and phosphorylation of AMPK, ACC, and HSL was increased in ASE and CGA treatment. This indicates that treatment with ASE and CGA inhibits lipid accumulation through the inhibition of fat synthesis, promotion of β-oxidation, and increased lipolysis (Figure 9).

## 4. Discussion

ROS are mainly produced during the oxidation reaction of the mitochondrial respiratory chain and are continuously generated during metabolic reactions in the body [48]. ROS are unstable and highly reactive molecules that easily react with other in vivo substances. They attack polymers in the body to degrade cell membranes and inhibit protein synthesis, leading to cytotoxicity and cancer [49,50]. Obesity involves fat accumulation via hyperplasia (increase in the number) and hypertrophy (increase in the size) of adipocytes. During lipid production, adipocytes produce various cytokines, ROS, and fatty acids that are responsible for metabolic syndromes such as diabetes, insulin resistance, and atherosclerosis [51,52]. Recent studies suggest that adipocyte ROS production influences lipid accumulation [9,10].

Flavonoids play an antioxidant role in regulating oxidative stress and have therapeutic and preventive effects on metabolic diseases such as cardiovascular disease, cancer and diabetes [10,53]. In this study, total phenolic content, flavonoid content, proanthocyanidin content, and ORAC values of ASE were investigated and the phenolic compounds were analyzed. The major bioactive compound of ASE was identified as chlorogenic acid (10.98 ± 0.04 mg/g). In various studies, it is known that *A. scaber* contains a lot of CQA, of which it is rich in chlorogenic acid [24,54].

The antioxidant activity of ASE was evaluated at the chemical and cellular levels, and the mechanism of inhibiting lipid accumulation through ROS inhibition in 3T3-L1 adipocytes was evaluated. G6PDH is a rate-limiting enzyme in the pentose phosphate pathway that increases the NADPH/NADP+ ratio, and NOX4 generates ROS in the process of transferring electrons from NADPH to oxygen [54,55]. Antioxidant enzymes present in adipocyte remove excess ROS [56]. SOD catalyzes superoxide with oxygen and hydrogen, and CAT, GPX, and GR use hydrogen peroxide to oxidize other substances and are known to prevent hydrogen peroxide accumulation in cells [57,58]. Our results showed that 3T3-L1 adipocytes inhibits ROS by downregulating ROS-producing G6PDH and NOX4 and increasing antioxidant enzymes such as SOD, catalase, GPx, and GR, which reduce excessive ROS production.

ROS produced by 3T3-L1 adipocytes promote adipogenesis and increase lipid accumulation. Lipid accumulation in adipocytes and ROS production are related to adipocyte differentiation. Differentiation of preadipocytes into mature adipocytes is regulated by increased insulin sensitivity and increased expression of C/EBPa and PPARγ [59]. C/EBPα and PPARγ increase the expression of aP2, which is expressed during adipocyte end-stage differentiation [60,61]. Recent studies have shown that an increase in intracellular ROS levels increases the expression of the C/EBPβ, a prerequisite regulator of C/EBPα and PPARγ expression, and influences the early differentiation stage of 3T3-L1 preadipocytes through regulation of mitotic clonal expansion during adipogenesis [10]. ASE reduced the levels of adipogenesis factors C/EBPα, PPARγ, and aP2, and regulated the levels of FAS, ACC, and HSL related to lipogenesis and lipolysis through AMPK activation. It was concluded that ASE has antioxidant and anti-obesity effects and may have therapeutic potential for use in various metabolic diseases caused by oxidative stress.

## Figures and Tables

**Figure 1 antioxidants-09-01290-f001:**
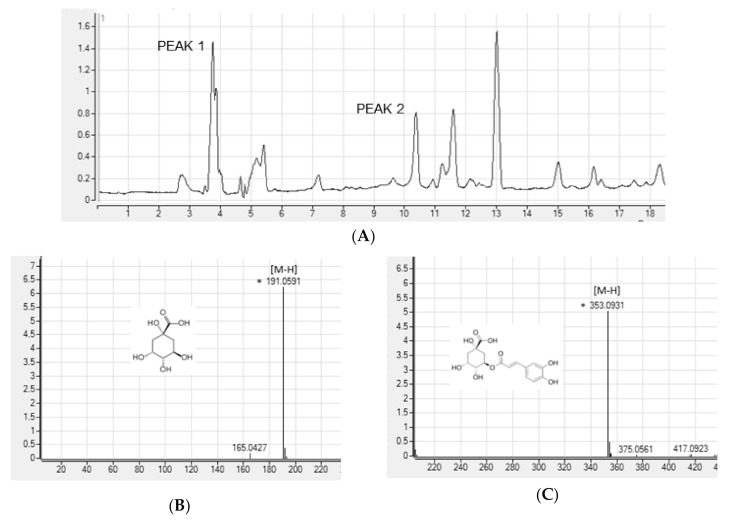
High-performance liquid chromatography (HPLC) chromatograms (**A**) and positive ion MS spectra (**B**,**C**) of *Aster scaber* ethanolic extract(ASE).

**Figure 2 antioxidants-09-01290-f002:**
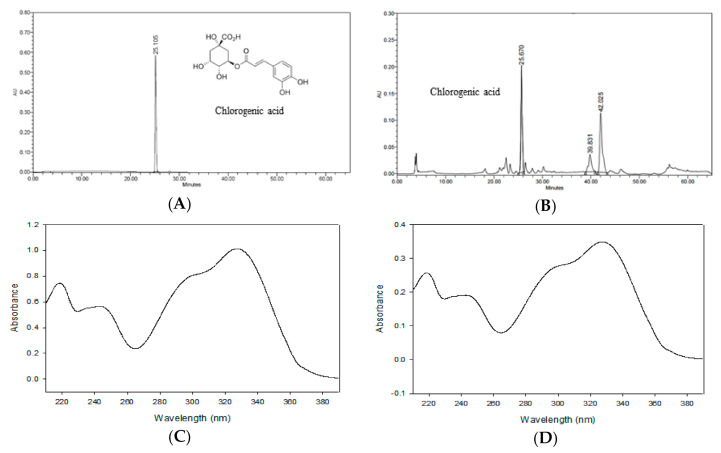
HPLC chromatograms of chlorogenic acid (**A**) and the ASE (**B**). Photodiode array (PDA) spectrum of chlorogenic acid (**C**) and ASE (**D**) from the chromatogram.

**Figure 3 antioxidants-09-01290-f003:**
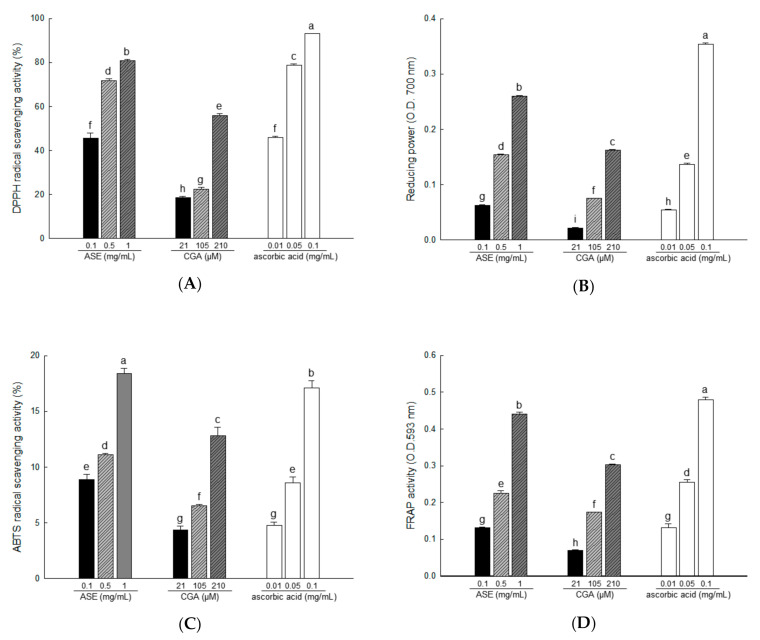
The antioxidant activity of ASE and chlorogenic acid (CGA) was evaluated through 2,2-diphenyl-1-picrylhydrazyl (DPPH) radical scavenging (**A**), reducing power (**B**), 2,2′-azino-bis(3-ethylbenzthiazoline-6-sulfonic acid) (ABTS) radical scavenging (**C**), and fluorescence recovery after photobleaching (FRAP) activities (**D**). Each value is expressed as the mean ± SD of triple determination. The other letters on the bar indicate a significant difference of *p* < 0.05 in the Duncan multi-range test. ^a,b,c,d,e,f,g,h and i^ significant differences among various parts of samples.

**Figure 4 antioxidants-09-01290-f004:**
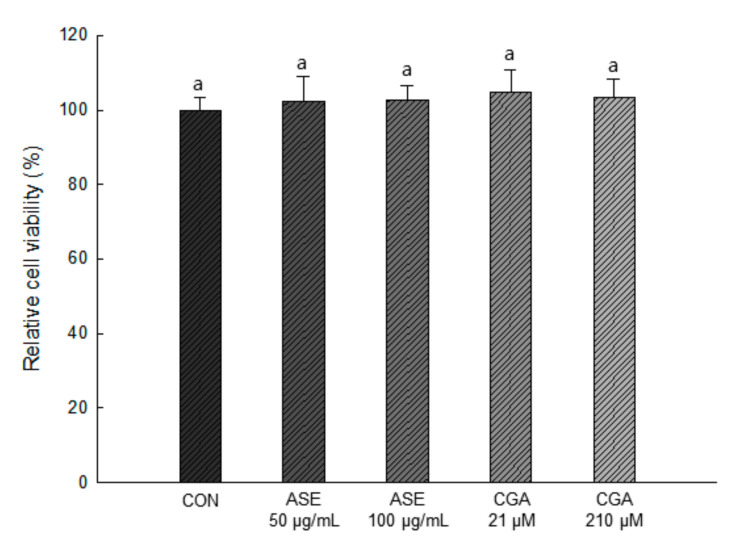
Effect of ASE and CGA on cell viability. After 6 days of MDI treatment, cell viability was measured by sodium 2,3,-bis(2-methoxy-4-nitro-5-sulfophenyl)-5-[(phenylamino)-carbonyl]-2H-tetrazolium (XTT) and N-methyl dibenzopyrazine methyl sulfate (PMS) reagents and absorbance was measured at 734 nm. Each value is expressed as the mean ± SD of triple determination. The other letters on the bars indicate a significant difference of *p* < 0.05 in the Duncan multi-range test. ^a^ significant differences among various parts of samples.

**Figure 5 antioxidants-09-01290-f005:**
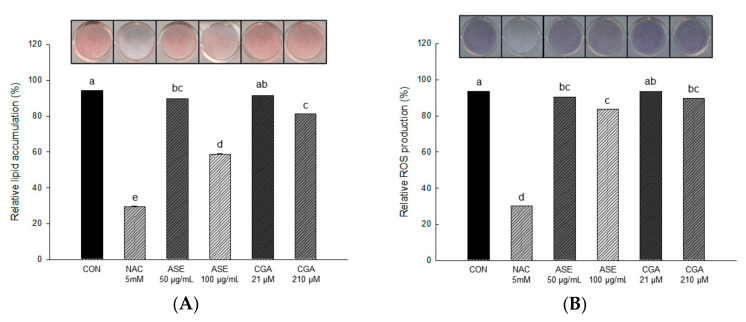
The effect of ASE and CGA on lipid accumulation (**A**) and reactive oxygen species (ROS) production (**B**) in 3T3-L1 adipocyte during the adipogenesis. After 6 days of MDI treatment, accumulated lipids were stained with oil red O (ORO) and absorbance was measured at 470 nm. ROS production was stained with nitroblue tetrazolium test (NBT) solution and absorbance was measured at 570 nm. Control group (CON), control cells which were differentiated with MDI; N-acetyl-L-cysteine (NAC), positive control cells which differentiated with MDI in the presence of NAC. Each value is expressed as the mean ± SD of triple determination. The other letters on the bars indicate a significant difference of *p* < 0.05 in the Duncan multi-range test. ^a, c, d, e, ab^ and ^bc^ significant differences among various parts of samples.

**Figure 6 antioxidants-09-01290-f006:**
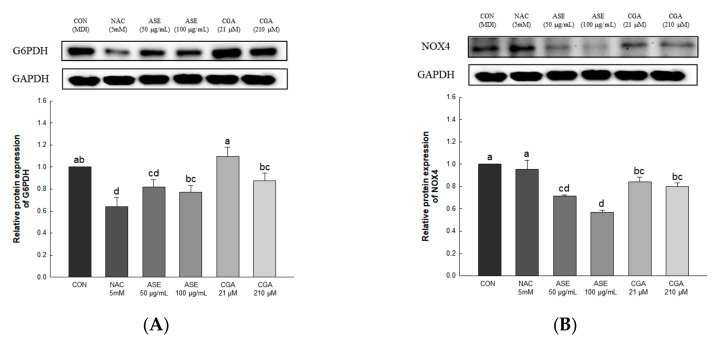
Effects of ASE and CGA on the expression level of ROS generating factor in 3T3-L1 adipocyte. Protein expression of G6PDH (A) and NOX4 (B). 3T3-L1 adipocytes differentiated for 6 days were treated with ASE (50 and 100 μg/mL) and CGA (21 and 210 μM). Protein expression was quantified through western blotting and normalized through GAPDH expression level. Each value is expressed as the mean ± SD of triple determination. The other letters on the bars indicate a significant difference of *p* < 0.05 in the Duncan multi-range test. ^a, d, ab, bc^ and ^cd^ significant differences among various parts of samples.

**Figure 7 antioxidants-09-01290-f007:**
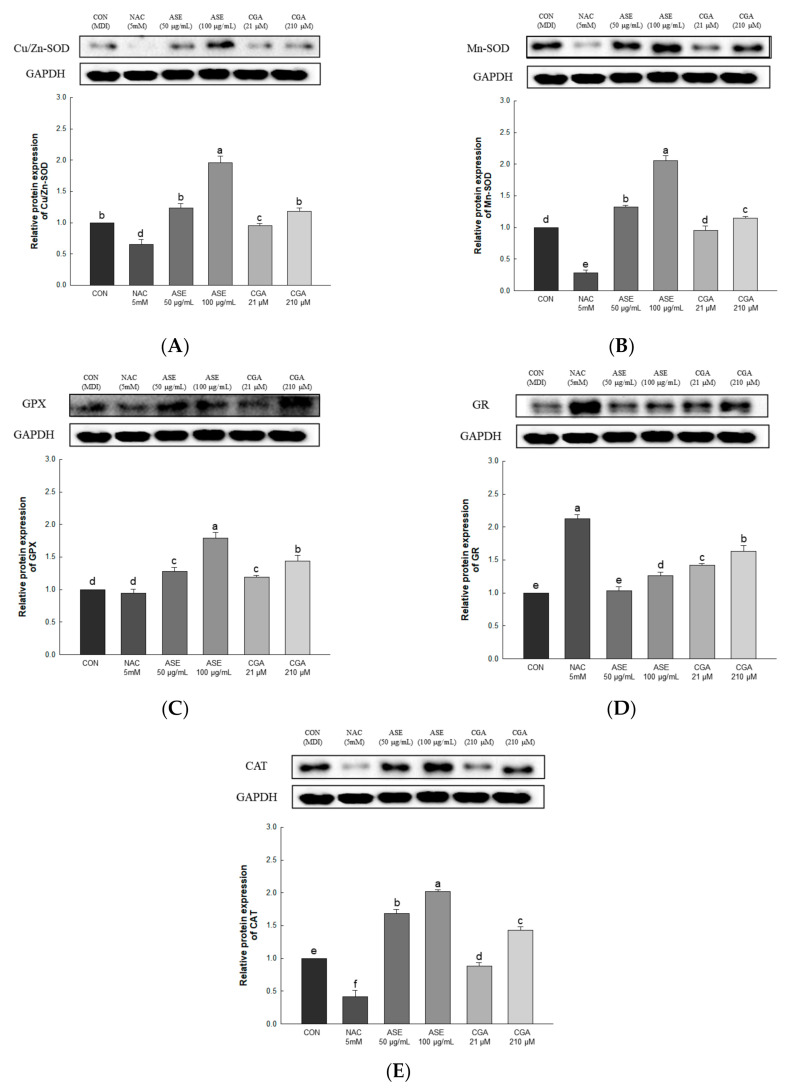
Effects of ASE and CGA on the expression level of ROS regulation antioxidant enzyme in 3T3-L1 adipocyte. Protein expression of Cu/Zn-SOD (**A**), Mn-SOD(**B**), GPX (**C**), GR (**D**) and CAT (**E**). 3T3-L1 adipocytes differentiated for 6 days were treated with ASE (50 and 100 μg/mL) and CGA (21 and 210 μM). Protein expression was quantified through western blotting and normalized through GAPDH expression level. Each value is expressed as the mean ± SD of triple determination. The other letters on the bars indicate a significant difference of *p* < 0.05 in the Duncan multi-range test. ^a–f^ significant differences among various parts of samples.

**Figure 8 antioxidants-09-01290-f008:**
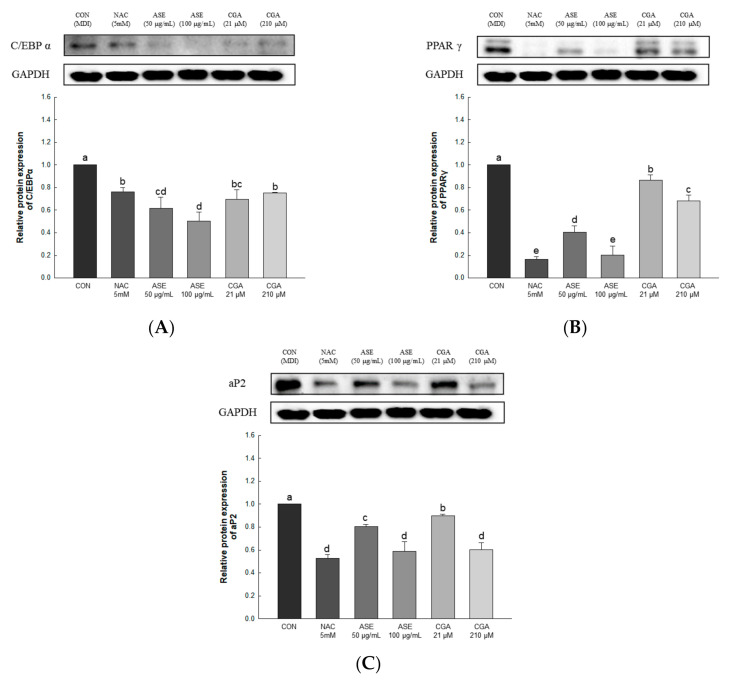
Effects of ASE and CGA on the expression level of adipogenic transcription factor in 3T3-L1 adipocyte. Protein expression of C/EBPα (**A**), PPARγ (**B**), and aP2 (**C**). 3T3-L1 adipocytes differentiated for 6 days were treated with ASE (50 and 100 μg/mL) and CGA (21 and 210 μM). Protein expression was quantified through western blotting and normalized through GAPDH expression level. Each value is expressed as the mean ± SD of triple determination. The other letters on the bars indicate a significant difference of *p* < 0.05 in the Duncan multi-range test. ^a, b, c, d, e, bc^ and ^cd^ significant differences among various parts of samples.

**Figure 9 antioxidants-09-01290-f009:**
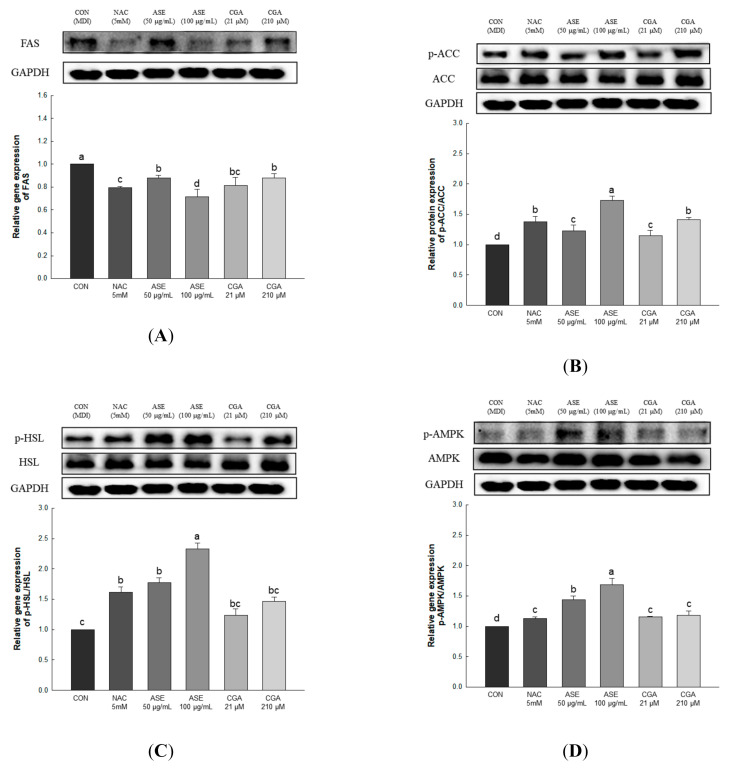
Effects of ASE and CGA on the expression level of AMP-activated protein kinase (AMPK) activation and lipid metabolism-related enzymes in 3T3-L1 adipocyte. Protein expression of FAS (**A**), p-ACC/ACC (**B**), p-HSL/HSL (**C**) and p-AMPK/AMPK (**D**). 3T3-L1 adipocytes differentiated for 6 days were treated with ASE (50 and 100 μg/mL) and CGA (21 and 210 μM). Protein expression was quantified through western blotting and normalized through GAPDH expression level. Each value is expressed as the mean ± SD of triple determination. The other letters on the bars indicate a significant difference of *p* < 0.05 in the Duncan multi-range test. ^a, b, c, d^, and ^bc^ significant differences among various parts of samples.

**Table 1 antioxidants-09-01290-t001:** Total phenol, flavonoid, proanthocyanidin contents, and oxygen radical absorbance capacity (ORAC) value of *Aster scaber* ethanolic extract (ASE).

Contents	*Aster scaber* Ethanolic Extract (ASE)
Total phenolic contents (mg GAE ^1^/g)	91.84 ± 0.52
Total flavonoid contents (mg QE ^2^/g)	53.39 ± 0.29
Total proanthocyanidin contents (mg CTE ^3^/g)	20.08 ± 0.84
ORAC value (μmol TE ^4^/g)	222.69 ± 0.30

^1^ Gallic acid equivalent, ^2^ quercetin equivalent, ^3^ catechin equivalent, ^4^ Trolox equivalent.

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
