# Peer review of "Radical Scavenging-Linked Anti-Adipogenic Activity of *Aster scaber* Ethanolic Extract and Its Bioactive Compound"

_antioxidants, 2020, doi:10.3390/antiox9121290_

Round 1

Reviewer 1 Report

The manuscript written by Choi et al. describes ethanolic extracts of a Korean herb in different aspects. Although the topic itself is quite classic, it contains novel aspects like polyphenol affects fat biosynthesis in adipose tissue. The introduction, experimental and results is written in a detailed way. The discussion summarise the results and the results are not discussed in respect to knowledge found in the literature. The reviewer find interesting to know if the described activities of ASE correspond to fractions which do not contain CGA.

Line 17-18: The sentence is difficult to understand.

Line 18: The reviewer does not understand why chlorogenic acid is mentioned.

Line 183: The used absorption wavelength is not selective enough. For the analysis of chlorogenic acid higher wavelength can be used which lead to higher selectivity. 210 nm is usually used when sensitivity of the method has to be increased.

Table 1-3: The reviewer suggest to add the tables into supplement. Several information were double written (table and text).

Line 269-270: Were the measured values comparable with the literature?

Section 3.2: Why only those two peaks were identified? There are other high intense peaks that might be also an interesting candidate. Please explain why did you concentrate on those two components? The other were not polyphenols? If yes, please provide this information in the text. Generally, polyphenols has specific UV-Vis spectra. This information could be used to filter out peaks that might belong to polyphenols. Did you do that?

Line 277: the identification was not described. Which parameters (calculation of elemental composition form isotope abundance distribution and exact mass, fragmentation pattern)? Did you use a database for the identification? These are not described only time aligned mass spectra were shown in Figure 1.

Section 3.2: CGA as bioactive compounds were selected according to the literature. Why did not you collect fractions and tested further?

Line 291-293: two different dimensions were used for comparison purpose. Please, use the same dimension to be able to directly compare the values.

Line 293: please define if fresh or dry weight were used (e.g. mg/g DW).

Table 5 is not needed.

Line 322 and Figure 3a: the reviewer suggest to use same dimension. Levels of CGA were defined in µM whereas of ASE and ascorbic acid in mg/ml.

Line 389-396: these information can be also read in the section of introduction.

Line 440: typo: point at the end of the sentence is doubled.

Author Response

The manuscript written by Choi et al. describes ethanolic extracts of a Korean herb in different aspects. Although the topic itself is quite classic, it contains novel aspects like polyphenol affects fat biosynthesis in adipose tissue. The introduction, experimental and results is written in a detailed way. The discussion summarise the results and the results are not discussed in respect to knowledge found in the literature. The reviewer find interesting to know if the described activities of ASE correspond to fractions which do not contain CGA.

Response: We appreciate the time and insightful comments of the reviewer. We have responded to each point brought up by the reviewer and also revised the manuscript to incorporate the reviewer’ suggestions using “providing more information”. We sincerely hope that reviewer will find our response satisfactory.

Point 1: Line 17-18: The sentence is difficult to understand.
Response 1: Sorry for being confused by the incorrect expression. Following your advice, the previously written content has been modified as follows. “Antioxidant activity was evaluated at the chemical level and the inhibition of lipid accumulation and reactive oxygen species (ROS) production was evaluated using 3T3-L1 adipocytes.”  Line 16-21 : “Antioxidant activity was measured at the chemical level through DPPH assay, reducing power assay, ABTS assay, and FRAP assay. In addition, it was measured in vitro through inhibition of ROS production in 3T3-L1 adipocyte, and inhibition of lipid accumulation was also evaluated.”

Point 2: Line 18: The reviewer does not understand why chlorogenic acid is mentioned.
Response 2: In this study, the antioxidant and anti-obesity effect of Aster scaber and chlorogenic acid selected as a bioactive compound of Aster scaber were evaluated. ASE and CGA are mentioned in line 18 because the mechanisms related to anti-obesity of ASE and CGA were evaluated. However, in Abstract, it was modified to remove the CGA of line 19-21.

Point 3: Line 183: The used absorption wavelength is not selective enough. For the analysis of chlorogenic acid higher wavelength can be used which lead to higher selectivity. 210 nm is usually used when sensitivity of the method has to be increased.
Response 3: Line 176, 181-183 : Sorry for being confused by the incorrect expression. For the analysis of chlorogenic acid, the analysis wavelength is 280 nm, and the analysis conditions were conducted according to previous studies. References have been added to the experimental method.

Point 4: Table 1-3: The reviewer suggest to add the tables into supplement. Several information were double written (table and text).
Response 4: Thank you for telling me the wrong notation. Table 1-3 has been modified to Table S1-3.

Point 5: Line 269-270: Were the measured values comparable with the literature?
Response 5: I really appreciate your advice. The following content has been added. Line 278-286 : “Lee et al [10] found that the total phenolic and flavonoid content of the Aster scaber methanol extract was 183.5 ± 4.0 mg GAE/g and 103.9 ± 3.4 mg RE/g, respectively. The ORAC value was 41,638 ± 282 μM TE/g. In Thiruvengadam et al [39], the total phenolic and flavonoid content in the Aster scaber methanol extract was 183.39 ± 5.59 mg/g and 3.12 ± 0.09 mg/g, respectively. The difference in content may be affected by the difference in the extraction solvent, the origin of the sample, and the harvest time.”

Point 6: Section 3.2: Why only those two peaks were identified? There are other high intense peaks that might be also an interesting candidate. Please explain why did you concentrate on those two components? The other were not polyphenols? If yes, please provide this information in the text. Generally, polyphenols has specific UV-Vis spectra. This information could be used to filter out peaks that might belong to polyphenols. Did you do that?
Response 6: We used a database (TCM, Tea, Metabolite Library; 2014, Agilant. USA) for identification. Matching peaks, N-Carbamylglutamate, quinic acid, chlorogenic acid, salsalate and monomethyl phthalate were identified in the database. Therefore, the peaks of the phenolic compounds quinic acid and chlorogenie acid were confirmed.

Point 7: Line 277: the identification was not described. Which parameters (calculation of elemental composition form isotope abundance distribution and exact mass, fragmentation pattern)? Did you use a database for the identification? These are not described only time aligned mass spectra were shown in Figure 1.
Response 7: We used a database (TCM, Tea, Metabolite Library; 2014, Agilent. USA) for identification. The peaks that match the database, quinic acid and chlorogenic acid, were identified and selected as candidates for bioactive compounds. The following content was added to the thesis. Line 290-294 : “To search for the bioactive compounds contained in ASE, phenolic compounds were analyzed using HPLC-Q/TOF-MS. We used a database (TCM, Tea, Metabolite Library; 2014, Agilent. USA) for identification. The peaks that match the database, quinic acid and chlorogenic acid. We analyzed the ion fragment data for two peaks with the highest area in the chromatogram (Figure 1, a). We estimated peak 1 as quinic acid (Figure 1, b) and peak 2 as chlorogenic acid (Figure 1, c).”

Point 8: Section 3.2: CGA as bioactive compounds were selected according to the literature. Why did not you collect fractions and tested further?
Response 8: I really appreciate your advice. We identified the bioactive compound of ASE by HPLC-MS and selected CGA through literature search. The separation of bioactive compounds from natural products is accomplished through extraction and fractionation. We extracted with 70% ethanol, which has a medium polarity of the solvent. If further research proceeds, the bioactive compound of ASE will be analyzed in depth through fractionation.

Point 9: Line 291-293: two different dimensions were used for comparison purpose. Please, use the same dimension to be able to directly compare the values.
Response 9: Line 296-298 : The unit of 26.37±0.11 μmol/g indicated in the reference was converted to mg/g. As a result, it was found to be 9.34±0.11 mg/g. We modified the word as you advised.

Point 10: Line 293: please define if fresh or dry weight were used (e.g. mg/g DW).
Response 10: Line 296-298 : The Aster scaber extract of reference was used after extraction and drying. Aster scaber extract was defined as mg/dry weight g. So, it was expressed as 9.343±0.11 mg/dry weight g.

Point 11: Table 5 is not needed.
Response 11: Thank you for telling me the wrong notation. Table 5 has been deleted.

Point 12: Line 322 and Figure 3a: the reviewer suggest to use same dimension. Levels of CGA were defined in µM whereas of ASE and ascorbic acid in mg/ml.
Response 12: The single component in the extract has a molecular weight. When 1g of substances A and B of the same weight exist, the number of substances A and B differs depending on the difference in molecular weight. Therefore, the CGA content (10.98mg/g) in 100 mg/mL of ASE was converted to molecular weight and determined to be 21 mM. CGA (21, 105, 210 μM) contained in ASE (0.1, 0.5, 1 mg/mL) used in the antioxidant experiment was used.

Point 13: Line 389-396: these information can be also read in the section of introduction.
Response 13: Following your advice, the previously written content has been modified as follows. Line 391-392 : "Antioxidant enzymes present in the cytoplasm and mitochondria remove excessively produced ROS [47]. Superoxide dismutase (Cu/Zn-SOD, Mn-SOD) present in the cytoplasm and mitochondria catalyze superoxide to oxygen and hydrogen peroxide, and catalase (CAT) is known to prevent accumulation of hydrogen peroxide in cells by oxidizing
other substances using hydrogen peroxide [48, 49]. Glutathione peroxidase (GPx) catalyzes the reaction to make water by reducing hydrogen peroxide, preventing the accumulation of hydrogen peroxide in cells. The antioxidant glutathione exists as reduced glutathione (GSH) and oxidized glutathione (GSSG), and glutathione reductase (GR) catalyzes the conversion of GSSG to GSH in the body to facilitate antioxidant activity [13, 14]."→"The expression levels of Cu/Zn-SOD, Mn-SOD, GPx, GR and CAT were evaluated to determine the effect of ASE and CGA treatment on ROS regulating antioxidant enzymes."

Point 14: Line 440: typo: point at the end of the sentence is doubled.
Response 14: Line 428-429 : Thank you for telling me the wrong notation. We modified the word as you advised.

Reviewer 2 Report

Page 1 line 27: there should be spaces after ;

Carefully check writing there are mistakes like CO2, starting sentence with small letter instead of capital letter

2.2. Total phenolic, flavonoid, anthocyanin contents and ORAC value; 3.1. Total phenolic, flavonoid, anthocyanin contents and ORAC value – there is no anthocyanin content in methods section but there is method for proanthocyanidins. Also results for anthocyanin content was presented through catechin equivalent what is not normally used.

Repetitions should be given in methods section.

Discussion should contain comparison of results with other studies.

Check reference guidelines.

Why authors choose ethanol extraction. It would be better if they compared different solvents as well, since different extraction efficiency could be achieved as they mentioned that there were some results on methanol extracts.

Round 2

Reviewer 2 Report

Author replyed on all comments and conducted nesecery modification thus manuscript can be accept

Author replied on all comments and conducted necessary modification thus manuscript can be accepted.